# Synergistic Effect of Abietic Acid with Oxacillin against Methicillin-Resistant *Staphylococcus pseudintermedius*

**DOI:** 10.3390/antibiotics10010080

**Published:** 2021-01-15

**Authors:** Elisabetta Buommino, Adriana Vollaro, Francesca P. Nocera, Francesca Lembo, Marina DellaGreca, Luisa De Martino, Maria R. Catania

**Affiliations:** 1Department of Pharmacy, University of Naples Federico II, 80131 Naples, Italy; frlembo@unina.it; 2Department of Molecular Medicine and Medical Biotechnology, University of Naples Federico II, 80131 Naples, Italy; vollaroadriana@libero.it (A.V.); mariarosaria.catania@unina.it (M.R.C.); 3Department of Veterinary Medicine and Animal Production, University of Naples Federico II, 80137 Naples, Italy; francescapaola.nocera@unina.it (F.P.N.); luisa.demartino@unina.it (L.D.M.); 4Department of Chemical Sciences, University of Naples Federico II, 80126 Naples, Italy; dellagre@unina.it

**Keywords:** abietic acid, *Staphylococcus pseudintermedius*, methicillin-resistant *Staphylococcus pseudintermedius*, methicillin-susceptible *Staphylococcus pseudintermedius*, antimicrobial activity, synergistic interaction

## Abstract

Resin acids are valued in traditional medicine for their antiseptic properties. Among these, abietic acid has been reported to be active against methicillin-resistant *Staphylococcus aureus* (MRSA) strains. In veterinary healthcare, the methicillin-resistant *Staphylococcus pseudintermedius* (MRSP) strain is an important reservoir of antibiotic resistance genes including *mecA*. The incidence of MRSP has been increasing, and treatment options in veterinary medicine are partial. Here, we investigated the antimicrobial and antibiofilm properties of abietic acid against three MRSP and two methicillin-susceptible *Staphylococcus pseudintermedius* (MSSP) strains, isolated from diseased pet animals and human wound samples. Abietic acid showed a significant minimal inhibitory concentration (MIC) value ranging from 32 to 64 μg/mL (MRSPs) and 8 μg/mL (MSSP). By checkerboard method we demonstrated that abietic acid increased oxacillin susceptibility of MRSP strains, thus showing a synergistic interaction with oxacillin. Abietic acid was also able to contrast the vitality of treated MSSP and MRSP1 biofilms at 20 μg/mL and 40 μg/mL, respectively. Finally, the compound moderately reduced *mecA*, *mecR1* and *mec1* gene expression. In conclusion, the results here reported demonstrate the antimicrobial activity of abietic acid against MRSP and support the use of this compound as a potential therapeutic agent to be used in combinatorial antibiotic therapy.

## 1. Introduction

*Staphylococcus pseudintermedius* is a normal inhabitant of the skin and mucosa of healthy dogs and cats. Over the last decade, *S. pseudintermedius* has emerged as a critically opportunistic animal pathogen causing primarily infections such as pyoderma and otitis [1]. The spread of methicillin-resistant *Staphylococcus pseudintermedius* (MRSP) worldwide represents a health problem for both companion animals and humans since zoonotic transmission has been documented [2,3]. The MRSP strains possess a mobile genetic element, the staphylococcal cassette chromosome (SCC*mec*), that include the *mec* gene (*mecA* and *mecC*) and its regulatory genes *mecR1* and *mec1*. Additionally, *mecB* has recently been detected in methicillin-resistant *Staphylococcus aureus* (MRSA) [4]. The *mec* gene encodes a modified PBP, namely the PBP2A, that presents a decreased binding affinity to beta-lactams, leading to ineffectiveness of antibiotic [4]. Since *mec* gene is located on a mobile element of the chromosome it can be easily transferred between staphylococcal species. The ability to form biofilm is one of the main virulence factors studied in bacteria [5]. Stefanetti et al. demonstrated that up to 96% of *S. pseudintermedius* strains isolated from canine infections were able to produce biofilm [6]. Biofilm formation can facilitate the colonization of different body sites in dogs and hinder the infection treatment. Walker et al. reported MIC values for biofilm-associated *S. pseudintermedius* significantly higher than for planktonic bacteria [7]. In another study, Meroni et al. reported that 68% of *S. pseudintermedius* isolates were able to form biofilms, but 100% of the multidrug resistant strains were slime producers; therefore, it seems that biofilm production can facilitate bacteria horizontal gene transfer [8]. Thus, discovery of novel agents for treatment of MRSP-associated and biofilm related infections is highly warranted.

To restore inefficacious critically important antibiotics, some research has been focused on the use of old drugs in synergic association with new antimicrobial, able to potentiate or restore their efficacy. The combination therapy is an important aspect of modern medicine in the design of new antimicrobials. The advantage relies in the possibility to repurpose existing, clinically-approved agents, accelerating the discovery and development of new therapies [9]. In this context, phytomedicals in combination with commercially available antibiotics represent an alternative.

Natural products from plants, fungi and bacteria have been successfully used in the past for their antimicrobial activity, but this unlimited source has only been partially investigated in the search of new antimicrobial agents [10,11,12,13]. Abietic acid is a diterpenoid with interesting antiviral, antibiotic and antifungal properties, whose main source is the resin of pine trees and other conifers [14,15,16]. Such resin is produced to defend the plant from the attack of insects or microorganisms in presence of tissue injury. This can in part explain the intrinsic activity of abietic acid against bacteria (mainly Gram-positive) and fungi [17].

Here, we first analyze the antimicrobial and antibiofilm properties of abietic acid against *S. pseudintermedius.* We also investigate the synergistic interaction between abietic acid and oxacillin against MRSP strains.

## 2. Results

### 2.1. Identification and Antimicrobial Susceptibility of S. pseudintermedius Strains

Two veterinary strains and three human strains of *S. pseudintermedius* were characterized for their pattern of antibiotic susceptibility, as reported in Table 1 [18]. One veterinary strain resulted oxacillin resistant (here named MRSP1), while the second strain resulted susceptible (MSSP). Concerning the strains isolated by human wounds two of the three resulted resistant to all the beta-lactams tested (MRSP2 and MRSP3). The third strain resulted susceptible to all antibiotics tested (Ctrl strain).

The phenotypic analysis of methicillin resistance was confirmed investigating by PCR the presence or absence of *mecA*, *mec1*, *mecR1* genes, as previously reported [19]. The results demonstrated that only one veterinary and two human strains possessed the *mec* operon (named MRSP1, MRSP2 and MRSP3, respectively), while the second veterinary strain did not (methicillin-susceptible *Staphylococcus pseudintermedius* (MSSP)) (data not shown).

### 2.2. Antimicrobial Activity of Abietic Acid

The antimicrobial activity of abietic acid was assessed against MRSP and MSSP strains. The MIC values are shown in Table 2. Abietic acid showed a significant MIC_90_ value at 8 μg/mL (Ctrl and MSSP), 32 μg/mL (MRSP1 and MRSP3) and 64 μg/mL (MRSP2). The time-kill assay results, presented in terms of the changes in the log10 CFU/mL of viable colonies, suggested a bacteriostatic activity for abietic acid (Figure 1) against all *S. pseudintermedius* strains. Abietic acid was able to inhibit bacterial growth already 1 h after the treatment, reducing the number of CFU by 45% for MSSP, 31% for MRSP1, 26% for MRSP2 and MRSP3, compared to untreated bacteria. The highest inhibition was observed 6 h after abietic acid exposure with a percentage of reduction by 78% for MSSP, MRSP2 and MRSP3, and about 74% for MRSP1. However, 24 h after cell growth resumption was observed in abietic acid-treated strains, even though not comparable to untreated cells (<3log_10_), thus confirming the bacteriostatic effect of abietic acid on all *S. pseudintermedius* strains.

### 2.3. Synergistic Study

The synergic interaction between abietic acid and oxacillin was determined by checkerboard assay. Oxacillin and abietic acid were used alone and in combination against MRSP planktonic cells. The drug concentration increased along the rows and column from 0 μg/mL to 32 μg/mL (abietic acid) and 0 μg/mL to 10 μg/mL (oxacillin). The highest synergistic interaction was obtained in the wells with the best combination of values, namely 4 μg/mL abietic acid (1/8 MIC) and 2.5 μg/mL oxacillin (1/4 MIC) for MRSP1 and MRSP3; 8 μg/mL abietic acid (1/8 MIC) and 2.5 μg/mL oxacillin (1/4 MIC) for MRSP2. The fractional inhibitory concentration (FIC) index, equal to 0.375, confirmed the synergistic effect of abietic acid and oxacillin (Table 2).

### 2.4. Molecular Analysis

We used RT-PCR to investigate the expression of *mecA, mec1* and *mecR1* genes and to evaluate the influence of abietic acid on their expression. Figure 2 shows the percentage of gene expression increase/decrease after treatment with abietic acid or oxacillin. Our data reported that 30 min treatment with subinhibitory concentration of oxacillin (1 μg/mL) strongly induced all the genes of *mec* operon in MRSP1, MRSP2 and MRSP3. On the contrary, subinhibitory concentrations of abietic acid (16 μg/mL for MRSP1 and MRSP3, and 32 μg/mL for MRSP2) moderately reduced the expression of *mecA, mec1* and *mecR1* genes.

### 2.5. Effect of Abietic Acid on Biofilm Formation

Crystal violet assay was used to test the ability of abietic acid to inhibit biofilm formation of treated MSSP and MRSP1. We tested sub-MIC concentrations ranging from 0.5 μg/mL to 2 μg/mL (1/16 MIC and 1/4 MIC for MSSP; and 1/64 MIC and 1/16 MIC for MRSP1, respectively) at which no effect on the planktonic growth was evident. Abietic acid at 2 μg/mL strongly reduced MSSP biofilm biomass, by causing 73% inhibition (Figure 3). The inhibition of MSSP biofilm formation was dose dependent and statistically different from the control for each concentration tested. On the other hand, abietic acid was not able to inhibit MRSP1 biofilm formation at any of the tested concentrations.

### 2.6. Effect of Abietic Acid on Mature Biofilm

The ability of abietic acid to destroy one-day-old biofilm of MSSP and MRSP1 was evaluated by crystal violet staining and XTT assay. Cristal violet assay showed that abietic acid did not exert a considerable effect on biofilm mass. It caused 50% reduction in the mass of the MSSP biofilm at all concentrations, while the mass of the MRSP biofilm was reduced by only 15% at the highest concentration tested (Figure 4 upper panel). On contrary, abietic acid showed a promising effect on biofilm viability. At 10 and 20 μg/mL, abietic acid was able to reduce the vitality of MSSP biofilm by 50% and 75%, respectively, while at 20 and 40 μg/mL abietic acid caused a reduction in the number of live cells in MRSP1 biofilm by 40% and 80%, respectively. (Figure 4 lower panel).

The effect of abietic acid on the mature biofilm of MSSP and MRSP1 was also observed by confocal microscopy at 20 μg/mL and 40 μg/mL, respectively. Images of treated biofilms of both strains showed a deep and compact red signal from the top layers to innermost. This means that cells within biofilm were killed by abietic acid (Figure 5B,D).

## 3. Discussion

In recent years, the high prevalence of methicillin-resistant staphylococcal species in veterinary medicine has become increasingly evident [20]. Among methicillin-resistant staphylococci (MRS), *S. pseudintermedius* (MRSP) is a pathogen of great importance not only in veterinary, but also in human medicine. However, infections caused by *S. pseudintermedius* in humans are often underreported due to inaccurate identification as *S. aureus*. The frequency to develop infections associated with colonization of MRSP in people daily in contact with animals is considerably increased. Rodrigues et al. [21] showed that 60% of participants to their study were found to be colonized by at least one MRS. Methicillin-resistance is caused by expression of modified penicillin-binding protein (PBP). *S. pseudintermedius* carrying *mecA* gene is not only resistant to penicillins, cephalosporines and carbapenemes, but often also possess resistance to macrolides, lincosamides and streptogramins. For this reason, the treatment options for MRSP infections have become very limited and represent a new challenge in veterinary medicine.

As a response to the imminent lack of effective antibiotics in veterinary field and the increase of zoonotic transmission, the antimicrobial properties of abietic acid were here investigated. Abietic acid and its derived compounds are already known for their antibacterial activity. Helfenstein et al. [22] reported an MIC value for abietic acid of 60 μg/mL against *S. aureus* and 8 μg/mL against methicillin-resistant *S. aureus, S. epidermidis* and *Streptococcus mitis*; in another study, an MIC value of 800 μg/mL against *S. epidermidis* was reported [23]. A recent review discussed about the promising role of antibiotic resistance breakers (ARB) classes of compounds, among which abietane diterpenes are enumerated with the efflux pumps inhibitors [24]. Among these, carnosic acid and carnosol, two diterpenes, at concentrations of 10 μg/mL induced two- and four-fold reductions in the MIC of tetracycline, respectively, against *S. aureus* strains containing Msr(A) and Tet(k) pumps [25].

Here, we demonstrated that abietic acid strongly reduced both MSSP and MRSP growth showing a bacteriostatic activity. Bacterial growth was inhibited already 1 h after, reaching the peak after 6 h of exposure. However, despite the growth resumption observed 24 h after, a slight decrease of cell growth was demonstrated, compared to control strain. Noteworthy, our results are strengthened by activity of abietic acid on the MRSP strains isolates from human samples. This result is encouraging, in consideration of high risk of zoonotic transmission of multiresistant strains.

To get a better understanding of antimicrobial activity of abietic acid against MRSP, we examined the interaction between abietic acid and oxacillin via the checkerboard method and described it in terms of FIC. The FIC index of abietic acid in combination with oxacillin was 0.375, which according to Pillai et al. [26] indicates a synergistic interaction. MIC of oxacillin for our MRSPs was 10 μg/mL, while if used in combination with abietic acid MIC value of oxacillin was reduced to 2.5 μg/mL. Although the value is still far from oxacillin breakpoint for *S. pseudintermedius* [27], this result is of great interest since we demonstrated that abietic acid was able to increase oxacillin susceptibility of MRSP strains. This could result from the ability of abietic acid to modulate *mec* operon. In line with the literature, here we showed that sub-inhibitory concentration of oxacillin strongly induced all the genes of the *mec* operon in the MRSP strains, while abietic acid treatment had the opposite effect [28]. However, we did not observe a strong inhibition of the genes included in the operon, but rather a mild repression. We can speculate that the reduced production of PBP2A enzyme in part abrogates the MRSP defense mechanism against beta-lactam antibiotics, thus partially restoring the activity of oxacillin. In fact, it has already been proposed that compounds having synergic effect with beta-lactams may inhibit the PBP2A activity [29,30]. In addition, the abietic acid activity here observed might be attributable to the carboxylic group, which interacts with the lipid component of the bacterial cellular membrane and allows the penetration of the molecule inside the membrane, altering its functions [23,31].

An important virulence factor of *S. pseudintermedius* is the ability to produce biofilm. In a study on 710 clinical isolates from healthy and diseased dogs, Little et al. showed that the proportion of biofilm-producing isolates from diseased dogs was significantly greater when compared with isolates from healthy ones [32]. Here we showed that at concentrations much lower than MIC values, abietic acid was able to inhibit by 73% MSSP biofilm formation, whereas it was unable to prevent biofilm formation by MRSP. At dose just above MIC, abietic acid induced a drastic reduction of preformed biofilm vitality of both MSSP and MRSP strains. Abietic acid efficacy on the viability of mature *Streptococcus mutans* biofilm was described by Ito et al., but at higher concentrations [33]. Although abietic acid had no effect on biofilm mass, we can hypothesize that due to its small size it penetrates biofilm matrix and kill the embedded bacteria. Further studies are needed to clarify the penetration and killing mechanisms. However, these findings might be of therapeutic interest because biofilm significantly affects antimicrobial susceptibility of both MSSP and MRSP from dogs [7]. Of interest, human *S. pseudintermedius* isolates also form structurally complex biofilm that are intrinsically resistant to antibiotics [34], thus complicating zoonotic infections.

In our opinion, a future use of abietic acid to fight bacterial infections is also supported by its proven antioxidant activity [23]. As known, the infectious process is associated to inflammatory response and the resulting oxidative stress is, undoubtedly, of importance in the mechanism defence of the host, but can cause tissue damage during the inflammation. In this context, the reported antioxidant properties of abietic acid might be protective during the infection for the host tissues. Worthy of note, abietic acid presented good biocompatibility as assessed by hemolytic assay, demonstrating a negligible lytic activity, and it was no toxic against human normal fibroblasts [23,35].

## 4. Materials and Methods

### 4.1. Chemical Used

Abietic acid (75%) (Alfa Aesar, Ward Hill, MA, USA) was dissolved in ethanol and diluted in Mueller–Hinton broth to give a stock solution. XTT (Roche Diagnostics, Monza, Italy); PBS, Gram staining, crystal violet-staining (Sigma-Aldrich, Milan, Italy); Penicillin G, ampicillin, amoxicillin-clavulanic acid and oxacillin, ceftriaxone, ceftazidime, imipenem, chloramphenicol, ciprofloxacin, gentamicin, streptomycin, clindamycin, erythromycin, vancomycin, tetracycline, linezolid, sulfametoxazole-trimethoprim (Oxoid Ltd., Basingstoke, UK).

### 4.2. Bacterial Strains and Culture

Five *S. pseudintermedius* strains were used in the study. Two strains were isolated from dogs with otitis externa at the Microbiology Laboratory of the Department of Veterinary Medicine and Animal Production, University of Naples Federico II (Italy). Three *S. pseudintermedius* strains were isolated by human wounds at the Bacteriology and Mycology Section of the Department of Molecular Medicine and Medical Biotechnology, School of Medicine and Surgery, University of Naples Federico II (Italy). The samples were plated on blood agar base supplemented with 5% sheep blood and on mannitol-salt agar and incubated aerobically at 37 °C for 24–48 h. Bacteria were firstly examined by macroscopic observation of the colonies, Gram staining, standard laboratory methodologies (catalase and staphylocoagulase tube test), and, finally, identified by matrix-assisted laser desorption ionization-time of flight mass spectrometry (MALDI-TOF MS) (Bruker Daltonics, Macerata, Italy).

### 4.3. Molecular Analysis

Genomic DNA extraction was performed by using GenUp Bacteria gDNA kit (BiotechRabbit, Berlin, Germany) according to the manufacturer’s instructions. All the isolates were tested for *S. pseudintermedius*-specific thermonuclease *nuc* gene and *mec* operon using the polymerase chain reaction (PCR) [36]. One µL of DNA was amplified in a reaction mixture containing 10 mM Tris–HCl (pH 8.3), 1.5 mM MgCl2, 50 mM KCl, 10 μM dNTP, 10 μM forward and reverse primers (*MecA* F-5′-TCCACCCTCAAACAGGTGAA-3′, R-5′-TGGAACTTGTTGAGCAGAGGT; *Mec1* F-5′-TCATCTGCAGAATGGGAAGTT, R-5′-TTGGACTCCAGTCCTTTTGC; *MecR1* F-5′-AGCACCGTTACTATCTGCACA, R-5′- AGAATAAGCTTGCTCCCGTTCA; rRNA16S F-5′- CGGTCCAGACTCCTACGGGAGGCAGCA, R-5′-GCGTGGACTACCAGGGTATCTAATCC) and 2.5 U of Taq DNA polymerase (BiotechRabbit, Berlin, Germany) in a final volume of 25 µL. The cycling conditions were as follows: preheating for 5 min at 95 °C, followed by 32 cycles of denaturation at 95 °C/30 s, annealing at 55 °C/40 s, extension at 72 °C/30 s and final extension for 5 min at 72 °C. The expected PCR products were 139 bp for *MecA*, 103 bp for *Mec1* and 142 bp for *MecR1* primers. The reaction was carried out in a DNA thermal cycler (MyCycler, Bio-Rad, Milan, Italy). The PCR products were analyzed by electrophoresis on 1.8% agarose gel in TBE and analyzed on a Gel Doc EZ System (Bio-Rad, Milan, Italy)

### 4.4. Antimicrobial Susceptibility Testing

The antimicrobial susceptibility patterns of isolated strains were determined by disk diffusion test on Mueller–Hinton agar (Oxoid Ltd., Basingstoke, UK). The inhibitory zone diameters obtained around the antibiotic disks were measured after incubation for 24 h at 37 °C and evaluated according to the Clinical and Laboratory Standards Institute (2013). Twenty-two commercial antibiotic discs (Oxoid Ltd., Basingstoke, UK) were tested. MIC of abietic acid was determined in Mueller–Hinton medium by the broth microdilution assay, according to the European Committee on Antimicrobial Susceptibility Testing. The compound was added to bacterial suspension in each well yielding a final cell concentration of 5 × 10^5^ CFU/mL and a final compound concentration ranging from 0.125 to 128 μg/mL. Negative control wells were set to contain bacteria in Mueller–Hinton broth plus the amount of ethanol used to dilute each compound. Positive controls included oxacillin (2 μg/mL) and vancomycin (2 μg/mL). Medium turbidity was measured by a microtiter plate reader (Tecan, Milan, Italy) at 595 nm. Minimum bactericidal concentration (MBC) was defined as the concentration that caused ≥3log_10_ reduction in colony count from the starting inoculum plated on TSA, incubated for 24 h at 37 °C.

### 4.5. Killing Rate

Time kill assay was carried out as previously described by Olajuyigbe et al. [37] with minor modifications. Bacterial suspension (10^5^ CFU/mL) was added to microplates along with abietic acid at the MIC value. Plates were incubated at 37 °C on an orbital shaker at 120 rpm. Viability assessments were performed at 0, 2, 4, 6 and 24 h by plating 0.1 mL undiluted and 10-fold serially diluted samples onto Mueller–Hinton plates in triplicate. After the overnight incubation at 37 °C, bacterial colonies were counted and compared with counts from control cultures.

### 4.6. Checkerboard Method

The interaction between abietic acid and oxacillin against MRSP strains was evaluated by the checkerboard method in 96-well microtiter plates containing Mueller–Hinton broth. Briefly, abietic acid and oxacillin were serially diluted along the y and x axes, respectively. The final concentration ranged from 1/16 to 1 × MIC for oxacillin and from 1/64 to 1 × MIC for abietic acid. The checkerboard plates were inoculated with bacteria at an approximate concentration of 10^5^ × CFU/mL and incubated at 37 °C for 24 h, following which bacterial growth was assessed visually and the turbidity measured by microplate reader at 595 nm. To evaluate the effect of the combination treatment, the FIC index for each combination was calculated as follows: FIC index = FIC of abietic acid + FIC of oxacillin, where FIC of abietic acid (or oxacillin) was defined as the ratio of MIC of abietic acid (or oxacillin) in combination and MIC of abietic acid (or oxacillin) alone. The FIC index values were interpreted as follows: ≤0.5, synergistic; >0.5 to ≤1.0, additive; >1.0 to ≤2.0, indifferent; and >2.0, antagonistic effects [26].

### 4.7. RNA Isolation and RT-PCR

MRSP strain was treated with sub-MIC concentration (16 μg/mL) of abietic acid for 30 min. Total RNA was isolated by using GenUp Total RNA kit (BiotechRabbit, Berlin, Germany) according to the manufacturer’s instructions. DNA contamination from the total RNA was removed by incubation with DNase I (RNase-free DNase Set, Qiagen, Hilden, Germany). Measuring A_260_/A_280_ nm ratio assessed the nucleic acid purity. To generate cDNA, total RNA was reverse transcribed by using (RevertUP II Reverse Transcriptase, BiotechRabbit, Berlin, Germany) into cDNA using random hexamer primers (Random hexamer, Roche Diagnostics, Monza, Italy) at 48 °C for 60 min according to the manufacturer’s instructions. RT-PCR was carried out using 2μl of cDNA. Quantification data were normalized to the reference gene for 16S rRNA gene and analyzed by Image Lab software 5.2.1 (Bio-Rad, Milan, Italy).

### 4.8. Effect of Abietic Acid on Biofilm Formation

Abietic acid was added at concentrations ranging from 0.25 μg/mL to 2 μg/mL to an equal volume of medium containing 10^6^ CFU/mL of MSSP and MRSP1. Control cells were grown in medium broth alone. After culturing for 24 h at 37 °C, the supernatant was gently removed, and the wells were rinsed with 200 μL of PBS. The biofilm biomass was measured by staining with 0.1% crystal violet-staining and the absorbance measured at 595 nm using a microtiter plate reader (Bio-Rad, Milan, Italy).

### 4.9. Effect of Abietic Acid against Preformed Biofilm

Biofilm was formed in 96-well microtiter plates. After 24 h planktonic cells were removed, and the wells were rinsed with PBS. Abietic acid was then added at the final concentration ranging from 10 μg/mL to 128 μg/mL and the plates were incubated for 24 h at 37 °C. Control biofilms were grown without abietic acid. At the end of the experiment crystal violet-staining was performed to assess biofilm biomass, as above described.

### 4.10. Quantitation of Metabolic Activity of Mature Biofilm

The XTT [2,3-bis(2-methyloxy-4-nitro-5-sulfophenyl)-2H-tetrazolium-5-carboxanilide] reduction assay was used to quantify the metabolic activity of preformed biofilms of MSSP and MRSP1. The assay was conducted as described by Vollaro et al. with some modifications [38]. After the treatment, XTT was added to each well and the plate incubated at 37 °C for 40 min in the dark. Changes in the absorbance of XTT, due to the reduction of the tetrazolium salt, were measured spectrophotometrically at 490 nm.

### 4.11. Confocal Laser Scanning Microscopy

Confocal laser scanning microscopy (CLSM) was used to illustrate the effect of abietic acid on viability and architecture of mature biofilms of MSSP and MRSP1. Cells were grown in chambered cover glass (μ Slide 4 well; ibidi GmbH, Gräfelfing, Germany) in a static condition. Abietic acid was added on a 1-day-old biofilm at 2 × MIC value for each strain. After 24 h, biofilms were rinsed and stained by using a LIVE/DEAD^®^ BacLight Bacteria Viability stains (Life Technologies, Monza, Italy). The images were observed using an LSM 700 inverted confocal laser-scanning microscope (Zeiss, Arese, Milano, Italy), using a 10× objective lens with the signal recorded in the green channel for Syto9 (excitation 488 nm, emission 500–525 nm) and in red channel for PI (excitation 500–550 nm, emission 610–650 nm).

### 4.12. Statistical Analysis

Each experiment was performed in triplicate and repeated three times on different days. Arithmetic means and standard deviations were calculated. Student’s *t*-test was used to determine statistical differences between group means.

## 5. Conclusions

The results discussed above and the selective action of abietic acid on bacterial cells rather than eukaryotic ones strongly support the use of this compound as potential therapeutic agent in combinatorial antibiotic therapy. On the base of the results here reported, we cannot affirm if abietic acid can affect other regulatory mechanisms involved in the antibiotic resistance. Nevertheless, the use of obsolete antibiotics and ARB compounds in combination, to enhance the action of the former, have an undoubted advantage. The possibility of synergy between abietic acid and oxacillin, the combination of different mode of action (interaction with bacterial cellular membrane, modulation of *mec* operon, reduction of cell viability in mature biofilm), in association to good biocompatibility and antioxidant properties makes abietic acid an intriguing compound for future studies.

## Figures and Tables

**Figure 1 antibiotics-10-00080-f001:**
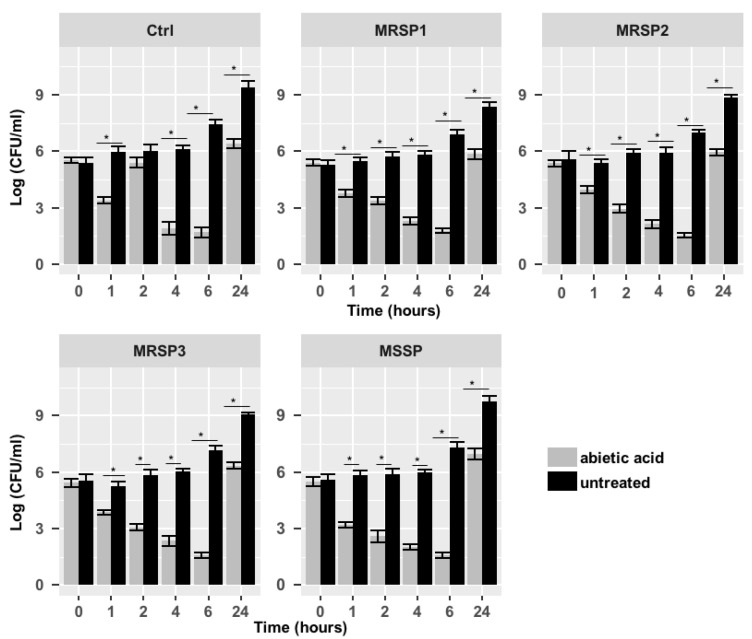
Time-kill assay of abietic acid against Ctrl, MRSP1-3 and MSSP strains. Each experiment is the result of three independent experiments performed in triplicate. Ctrl: human antibiotics susceptible *S. pseudintermedius* strain; MSSP: veterinary methicillin-susceptible strain; MRSP1: veterinary methicillin-resistant *S. pseudintermedius* strain; MRSP2 and 3: human methicillin-resistant *S. pseudintermedius* strains. * represents *p*  <  0.05.

**Figure 2 antibiotics-10-00080-f002:**
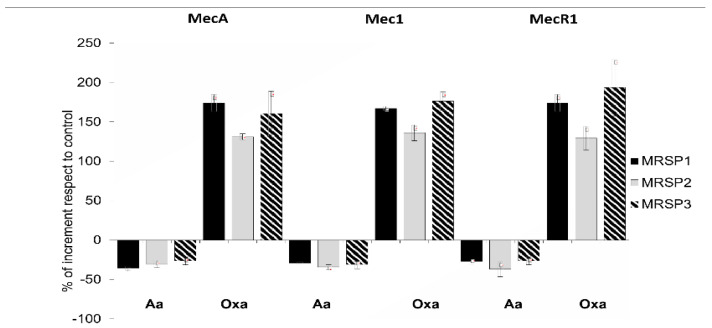
Relative expression of genes of *mec* operon in MRSP strains treated with subinhibitory concentration of abietic acid (16 mg/mL for MRSP1 and MRSP3, and 32 mg/mL for MRSP2) and 1 mg/mL oxacillin. Values represent the mean ± SD for three independent experiments. MRSP1: veterinary methicillin-resistant *S. pseudintermedius* strain; MRSP2 and 3: human methicillin-resistant *S. pseudintermedius* strains. Aa: abietic acid; Oxa: oxacillin.

**Figure 3 antibiotics-10-00080-f003:**
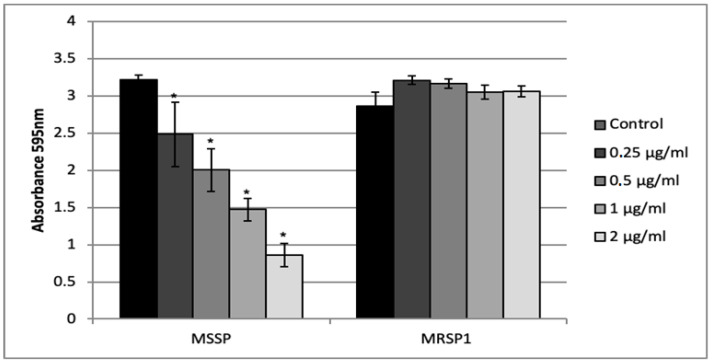
Activity of the abietic acid on *S. pseudintermedius* biofilm formation. Values represent the mean ± SD for three independent experiments. * indicates statistically significant difference (*p* < 0.05).

**Figure 4 antibiotics-10-00080-f004:**
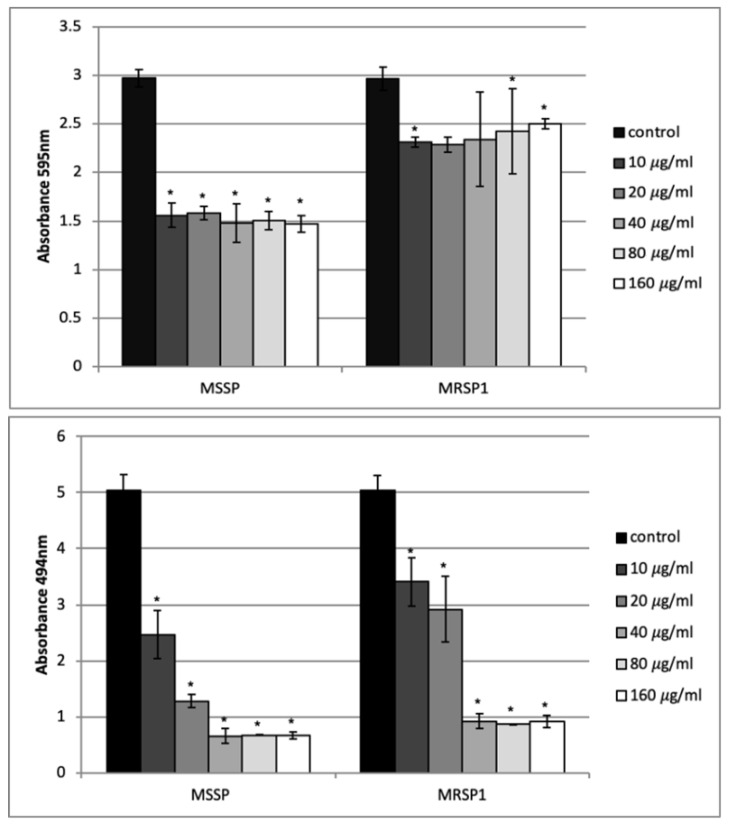
Effect of abietic acid on one-day-old biofilms of *S. pseudintermedius*. Upper panel: biofilm biomass obtained by crystal violet staining; bottom panel: biofilm vitality obtained by XTT assay. Bars represent the average values ± SD from three independent experiments. * represents *p* < 0.05.

**Figure 5 antibiotics-10-00080-f005:**
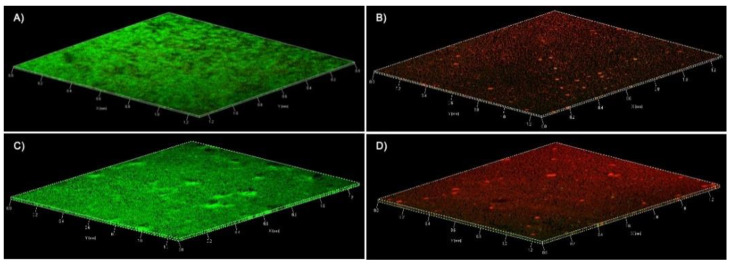
Confocal laser scanner microscopy micrographs of MSSP and MRSP1 biofilms. The panels contain three-dimensional images, green and red fluorescence is associated with live and dead cells, respectively. (**A**) MSSP untreated biofilm (**B**) MSSP biofilm treated with abietic acid at 20 µg/mL (**C**) MRSP1 untreated biofilm (**D**) MRSP1 biofilm treated with abietic acid at 40 µg/mL.

**Table 1 antibiotics-10-00080-t001:** List of antibiotics used and pattern of resistance. Penicillin G (P), ampicillin (AM), amoxicillin-clavulanic acid (AMC), oxacillin (OX), ceftazidime (CAZ), imipenem (IPM), chloramphenicol (C), ciprofloxacin (CIP), gentamicin (GM), streptomycin (S), clindamycin (CM), erythromycin (E), vancomycin (VA), tetracycline (TE), linezolid (LZD), sulfametoxazole-trimethoprim (SXT). S = susceptible; R = resistant.

Strains	P	AM	AMC	OX	CAZ	IPM	C	CIP	GM	S	CM	E	VA	TE	LZD	SXT
Ctrl	S	S	S	S	S	S	S	S	S	S	S	S	S	S	S	S
MSSP	R	R	S	S	S	S	S	S	S	S	S	S	S	R	S	S
MRSP1	R	R	R	R	R	R	S	R	R	R	S	R	S	R	S	R
MRSP2	R	R	R	R	R	R	S	S	S	S	S	R	S	R	S	R
MRSP3	R	R	R	R	R	R	S	S	R	R	S	S	S	R	S	S

**Table 2 antibiotics-10-00080-t002:** Minimal inhibitory concentration (MIC90), minimal bactericidal concentration (MBC) and fractional inhibitory concentration (FIC_index_) values of abietic acid on selected bacterial strains. n.d. = not determined. Each result is representative of three independent experiments performed in triplicate.

Strains	MIC (μg/mL)	MBC (μg/mL)	Oxacillin (μg/mL)	Vancomicin (μg/mL)	FIC_index_
Ctrl	8	16	<0.25	1	n.d.
MSSP	8	32	<0.25	2	n.d.
MRSP1	32	64	10	2	0.375
MRSP2	64	64	10	2	0.375
MRSP3	32	64	10	2	0.375

## Data Availability

Data is contained within the article.

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
