# Peer review of "Synergistic Effect of Abietic Acid with Oxacillin against Methicillin-Resistant Staphylococcus pseudintermedius"

_antibiotics, 2021, doi:10.3390/antibiotics10010080_

Round 1

Reviewer 1 Report

In the manuscript “Synergistic effect of abietic acid with oxacillin against methicillin-resistant Staphylococcus pseudintermedius” the authors investigated the antimicrobial and antibiofilm properties of abietic acid against three methicillin-resistant Staphylococcus pseudintermedius and two methicillin-susceptible Staphylococcus pseudintermedius strains, isolated from diseased pet animals and human wounds samples.

Major comments

  1. Introduction section is short. Rational of synergistic effect of natural compounds/products with antimicrobial agents/antibiotics against drug resistant bacterial pathogens should be added in introduction section.
  2. In section 2.1 authors have simply describe that isolated strains were resistant to oxacillin and all the antibiotics tested. It would better to describe those data in details by incorporating table and listing the name of all antimicrobial tested and their resistant or sensitivity pattern.
  3. Authors have described in discussion section that Abietic acid and its derived compounds are already known for their antibacterial activity. However, Abietic acid has also explored as antibiofilm agents against Streptococcus mutans elsewhere. Therefore, those finding should also be incorporated in introduction or discussion section.

Minor comments

  1. Presented paper needs English proofreading. Please, go thoroughly through the text and correct the grammar.
  2. Line #29, MRSP and MSSP should not be in abbreviated form in key words.
  3. Line #63 and #79, #85, #89, #90, #91, #118, #119, #130, #143, S. pseudintermedius should be in italic.
  4. Rewrite sentence in Line #107-#108: We used RT-PCR to investigate the expression……..……………..possible mechanism of action.
  5. The typo errors throughout the manuscript should be corrected.

Author Response

Dear Reviewer,

please find here point-by-point response to your suggestions. All the corrections were marked in red in the new manuscript. English has been improved. We hope that the revised version of our manuscript could be suitable for publication in the journal Antibiotics.

Major comments

  1. Introduction section is short. Rational of synergistic effect of natural compounds/products with antimicrobial agents/antibiotics against drug resistant bacterial pathogens should be added in introduction section.

We agree with your suggestion. We added a new paragraph in the introduction section regarding the synergism (Lines 55-60). A new reference has been added (Ref. n°9. Twarog, N.R.; Connelly, M.; Shelat, A.A. A critical evaluation of methods to interpret drug combinations. Sci Rep 2020, 105144, https://doi.org/10.1038/s41598-020-61923-1).

  1. In section 2.1 authors have simply describe that isolated strains were resistant to oxacillin and all the antibiotics tested. It would better to describe those data in details by incorporating table and listing the name of all antimicrobial tested and their resistant or sensitivity pattern.

Detailed description of resistance pattern has been introduced for each strain (section 2.1 lines 75-79, new table 1).

  1. Authors have described in discussion section that Abietic acid and its derived compounds are already known for their antibacterial activity. However, Abietic acid has also explored as antibiofilm agents against Streptococcus mutans elsewhere. Therefore, those finding should also be incorporated in introduction or discussion section.

This aspect has been already introduced in discussion section in the first submitted version of our manuscript (lines 228-229, ref 33).

Minor comments

  1. Presented paper needs English proofreading. Please, go thoroughly through the text and correct the grammar.

The manuscript has been thoroughly proofread and grammar corrected.

  1. Line #29, MRSP and MSSP should not be in abbreviated form in key words.

Modified

  1. Line #63 and #79, #85, #89, #90, #91, #118, #119, #130, #143, S. pseudintermedius should be in italic.

Sorry for this. Maybe some italics were lost in the final document.

  1. Rewrite sentence in Line #107-#108: We used RT-PCR to investigate the expression……..……………..possible mechanism of action.

We eliminate the sentence since it was not necessary in the paragraph 2.4.

  1. The typo errors throughout the manuscript should be corrected.

We have checked for errors in the manuscript and corrected it.

Reviewer 2 Report

 Overall the concept of this study seems to be important to resolve the problem of antibiotics resistance. This reviewer read the manuscript interesting. Each result was described simply and specifically. The potential of abietic acid seems to be so profound, the author's further study will be hopeful.

 So far, the results of this manuscript are of importance for the fundamental information. In that sense, this manuscript by Buommino et al. is suitable to publish for this journal.

Miner comment:

In the results, the name of bacteria should be italic style. Please correct.

L101~: The footnote of this table should be located below the table. Please move.

Author Response

Dear Reviewer,

please find here point-by-point response to your suggestions. All the corrections were marked in red in the new manuscript. English has been improved. We hope that the revised version of our manuscript could be suitable for publication in the journal Antibiotics.

  1. In the results, the name of bacteria should be italic style. Please correct.

Sorry for this. Maybe some italics were lost in the final document.

  1. L101~: The footnote of this table should be located below the table. Please move.

Done.

Reviewer 3 Report

The manuscript 'Synergistic effect of abietic acid with oxacillin against methicillin-resistant Staphylococcus pseudintermedius' reports interestings findings and it is quite well written. It could be published in Antibiotics once that a few concerns are fixed:

Quality of presentation:

There are a few of concerns related to the presentation of a scientific article:

1) The name of the bacteria should be always in italics (Staphylococcus pseudintermedius). Examples: in lines 63, 79, 85, 90... Please correct in all manuscript.

2) Line 56: please add a space here to separate 'Such' from the squared bracket: '].Such'

3) Like the name of the bacteria, the name of the genes should be always in italics (mecA, mec1, mecR1. Examples: in lines 70, 71, 107, 253... Please correct in all manuscript.

Minor concerns:

4) The abbreviation 'MRSMA' (line 40) looks like incorrect, should it be MRSA. If not please explain what means the second 'M'.

5) Please, in lines 65-71, insert the abbreviations MRSP1, MRSP2 and MRSP3 after the first citation (the descriptions of to which antibiotics each strain is sensititive/resistant). Looks like they are cited in order, but it would be more clear for the reader.

6) MIC and FIC abbreviations should be defined after its first appearance. I am aware that FIC is defined in material and methods, but at least explaining the abbreviation in line 99 is necessary.

7) Line 123: the expression 1/16 MIC and 1/4 MIC is appropriate for MSSP, whose MIC is 8 micrograms/mL (considering the range); but not for MRSP1 (32 micrograms/mL), for which it should be 1/64 MIC and 1/16 MIC. Please rewrite accordingly.

8) Figure 5 should have the legend in the same page that the body of the figure. Please fix to achieve it, for example, introducing the first paragraph of the discussion here and moving figure for next page.

9) It is necessary to indicate the name and vendors of all chemicals used (Abietic acid, XTT, PBS, oxacillin, vancomicin and any other chemical that I may have missed in this list) in a separate subsection in Materials and Methods. I suggest to rename section '4.1 Abietic acid' to '4.1. Chemicals used' and including all there. 

10) What does it mean the abbreviation ARB in line 341? Please define.

English:

English is good and readable although a careful revision would be advisable, to detect and fix more small mistakes as for example:

11) Line 128: If sentence is in negative ('was not able...') it should be 'at any of the tested concentrations' instead of 'at all tested concentrations'.

Author Response

Dear Reviewer,

please find here point-by-point response to your suggestions. All the corrections were marked in red in the new manuscript. English has been improved. We hope that the revised version of our manuscript could be suitable for publication in the journal Antibiotics.

1) The name of the bacteria should be always in italics (Staphylococcus pseudintermedius). Examples: in lines 63, 79, 85, 90... Please correct in all manuscript.

Sorry for this. Maybe some italics were lost in the final document.

2) Line 56: please add a space here to separate 'Such' from the squared bracket: '].Such'

Done.

3) Like the name of the bacteria, the name of the genes should be always in italics (mecA, mec1, mecR1. Examples: in lines 70, 71, 107, 253... Please correct in all manuscript.

Sorry for this. Maybe some italics were lost in the final document

Minor concerns:

4) The abbreviation 'MRSMA' (line 40) looks like incorrect, should it be MRSA. If not please explain what means the second 'M'.

We have corrected the abbreviation.

5) Please, in lines 65-71, insert the abbreviations MRSP1, MRSP2 and MRSP3 after the first citation (the descriptions of to which antibiotics each strain is sensititive/resistant). Looks like they are cited in order, but it would be more clear for the reader.

Paragraph 2.1. We have modified the sentence as suggested by the reviewer and also added a new table (Table 1 in the revised version).

6) MIC and FIC abbreviations should be defined after its first appearance. I am aware that FIC is defined in material and methods, but at least explaining the abbreviation in line 99 is necessary.

We agree with your suggestion and added the definition of MIC and FIC at their first appearance (lines 21-22; line 123).

7) Line 123: the expression 1/16 MIC and 1/4 MIC is appropriate for MSSP, whose MIC is 8 micrograms/mL (considering the range); but not for MRSP1 (32 micrograms/mL), for which it should be 1/64 MIC and 1/16 MIC. Please rewrite accordingly.

The reviewer is right. We have modified the sentence (line 142).

8) Figure 5 should have the legend in the same page that the body of the figure. Please fix to achieve it, for example, introducing the first paragraph of the discussion here and moving figure for next page.

We have modified as suggested.

9) It is necessary to indicate the name and vendors of all chemicals used (Abietic acid, XTT, PBS, oxacillin, vancomicin and any other chemical that I may have missed in this list) in a separate subsection in Materials and Methods. I suggest to rename section '4.1 Abietic acid' to '4.1. Chemicals used' and including all there. 

A list of all chemicals used has been added in the section 4.1 renamed “Chemicals used”.

10) What does it mean the abbreviation ARB in line 341? Please define.

The acronym ARB has been already defined in the discussion section, line 194.

English:

English is good and readable although a careful revision would be advisable, to detect and fix more small mistakes as for example:

11) Line 128: If sentence is in negative ('was not able...') it should be 'at any of the tested concentrations' instead of 'at all tested concentrations'

We have proofread the manuscript for possible errors.

Round 2

Reviewer 1 Report

  1. Legend of table 1 and table 2 should be above the table not below the table.
  2. It would better to explain 'S' and 'R' in table 1 legend.

Author Response

Dear Reviewer we have modified the manuscript as requested.

1) Legends to table 1 and 2 have been moved above the respective tables.

2) "S" and "R" have been explained in legend of table 1.